# The Persian plateau served as hub for *Homo sapiens* after the main out of Africa dispersal

Leonardo Vallini [1] ✉, Carlo Zampieri[1], Mohamed Javad Shoaee[2], Eugenio Bortolini [3], Giulia Marciani [3,4], Serena Aneli [5], Telmo Pievani[1], Stefano Benazzi[3], Alberto Barausse[1,6], Massimo Mezzavilla [1], Michael D. Petraglia [7,8,9] & Luca Pagani [1,10] ✉

A combination of evidence, based on genetic, fossil and archaeological findings, indicates that *Homo sapiens* spread out of Africa between ~70-60 thousand years ago (kya). However, it appears that once outside of Africa, human populations did not expand across all of Eurasia until ~45 kya. The geographic whereabouts of these early settlers in the timeframe between ~70-60 to 45 kya has been difficult to reconcile. Here we combine genetic evidence and palaeoecological models to infer the geographic location that acted as the Hub for our species during the early phases of colonisation of Eurasia. Leveraging on available genomic evidence we show that populations from the Persian Plateau carry an ancestry component that closely matches the population that settled the Hub outside Africa. With the paleoclimatic data available to date, we built ecological models showing that the Persian Plateau was suitable for human occupation and that it could sustain a larger population compared to other West Asian regions, strengthening this claim.

A growing body of evidence indicates that the colonisation of Eurasia by *Homo sapiens* was not a simple process, as fossil and archaeological findings support a model of multiple migrations Out of Africa from the late Middle Pleistocene and across the Late Pleistocene[1–6]. Traces of these early dispersals are also evidenced in the genome of our Neanderthal relatives, which illustrate interbreeding events as humans moved into Eurasia[7–10]. Early dispersals of our species were likely accompanied by population contractions and extinctions, though succeeded by a subsequent, large-scale wave at ~70–60 kya[11–15], from which all modern-day non-Africans descend[16,17].

The geographically widespread and stable colonisation of Eurasia appears to have occurred at ~45 kya through multiple population expansions associated with a variety of stone tool technologies[18,19]. Earlier incursions into Europe have been recorded[20–23], however, they failed to leave a significant contribution to later populations. A chronological gap of ~20 ky between the Out of Africa migration (~70–60 kya) and the stable colonisation (~45 kya) of West and East Eurasia can be identified, for which the geographic location and genetic features of this population are poorly known. On the basis of genetic and archaeological evidence, it has been suggested that the Eurasian population that formed the first stable deme outside Africa after ~70–60 kya can be characterised as a Hub population[18], from which multiple population waves emanated to colonise Eurasia, which would have had distinct chronological, genetic and cultural characteristics. It has also been surmised that the Hub population cannot be seen as simply the stem from which East and West Eurasians diverged. Instead, this was a more complex scenario, encompassing multiple expansions and local extinctions[18]. Previous studies, however,

[1]Department of Biology, University of Padova, Padova, Italy. [2]Department of Archaeology, Max Planck Institute for Geoanthropology, Jena, Germany. [3]Department of Cultural Heritage, University of Bologna, Bologna, Italy. [4]Research Unit Prehistory and Anthropology, Department of Physical Sciences, Earth and Environment, University of Siena, Siena, Italy. [5]Department of Public Health Sciences and Pediatrics, University of Turin, Turin, Italy. [6]Department of Industrial Engineering, University of Padova, Padova, Italy. [7]Human Origins Program, Smithsonian Institution, Washington, DC 20560, USA. [8]School of Social Science, The University of Queensland, Brisbane, QLD, Australia. [9]Australian Research Centre for Human Evolution, Griffith University, Brisbane, QLD, Australia. [10]Institute of Genomics, University of Tartu, Tartu, Estonia. ✉e-mail: leo.vallini.lv@gmail.com; luca.pagani@unipd.it

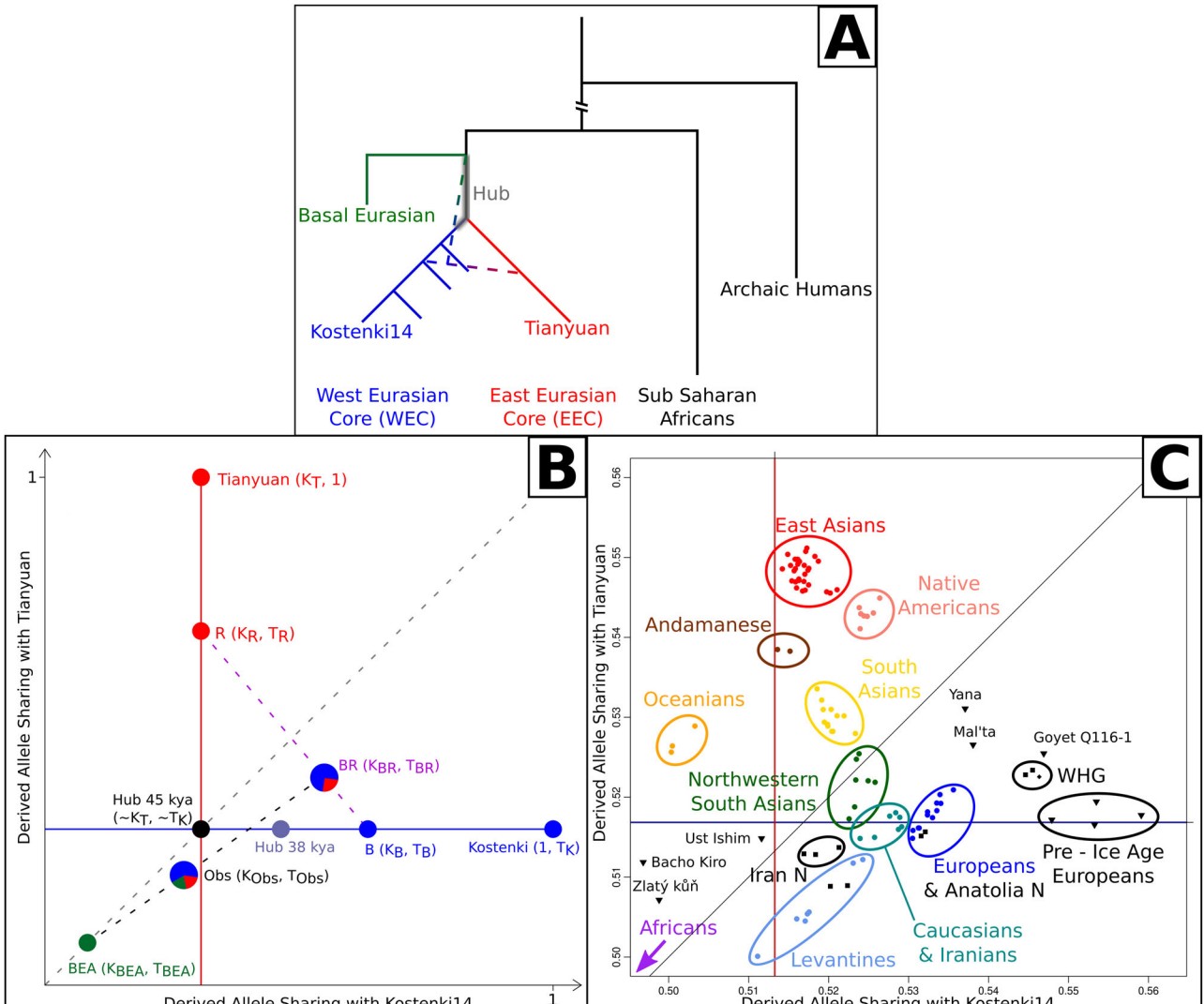

**Fig. 1 | Relationship and legacy of the West and East Eurasian Core populations.** Summary of the major population events (**A**) and schematic representation of our reference space and expected position of admixed and unadmixed populations (**B** – note that this panel is a rotation of the blue and red portion of **A**); derived allele sharing of each ancient or modern individual/population with Kostenki14 and Tianyuan (**C**); East Asians in red, Oceanians in orange, Native Americans in pink, South Asians in yellow, Northern South Asians in green, West Eurasians in blue, Levantines in cornflowerblue, ancient samples in black (N stands for Neolithic, WHG for Western Hunter Gatherers). A graph with all populations analysed, and individual population names are in Supplementary Fig. 1, Source Data in Supplementary Data 3.

have failed to delve into the potential geographic location of this Hub population[24], the overall scarcity of fossil evidence of *Homo sapiens* between 60 and 45 kya anywhere across Eurasia.

The aforementioned scenario was grounded in evidence stemming from ancient genomes from West and Central Eurasia[25,26] and China[27], indicating that the ancestors of present-day East Eurasians emerged from the Hub at ~45 kya (Fig. 1A, red branch). These emergent groups subsequently colonised most of Eurasia and Oceania, though these populations became largely extinct and were assimilated in West Eurasia[28] by a more recent expansion that took place by ~38 kya (Fig. 1A, blue branch). The first of these two expansions, whose associated ancestry we name here the East Eurasian Core (EEC), left descendants in Bacho Kiro, Tianyuan, and most present-day East Asians and Oceanians. The second expansion, which we name the West Eurasian Core (WEC), left descendants in Kostenki14, Sunghir, and subsequent West Eurasians, and in the genome of palaeolithic Siberians[29]. Crucially, the Hub population accumulated some drift together with the WEC in the millennia that elapsed between the EEC and the WEC expansions (Fig. 1A, grey area). Despite its key role during the peopling

of Eurasia, the geographic location and the genetic characteristics of the Hub population, remain obscure[24]. The outlined scenario is complicated by the need to account for the Basal Eurasian population (Fig. 1A, green), a group[30] that split from other Eurasians soon after the main Out of Africa expansion, hence also before the split between East and West Eurasians. This population was isolated from other Eurasians and later on, starting from at least ~25 kya[31,32], admixed with populations from the Middle East. Their ancestry was subsequently carried by the population expansions associated with the Neolithic revolution to all of West Eurasia.

Given the current impossibility of directly inferring the homeland of the Hub population from fossil remains, here we combine available genetic evidence (including both ancient and present-day genomes) and palaeoecological models to infer the geographic region that acted as a Hub for the ancestors of all present-day non-Africans during the initial colonisation of Eurasia. With our work, we show that populations from the Persian Plateau carry an ancestry component that closely matches the population that settled the Hub outside Africa, therefore pointing to the Persian Plateau as suitable for human occupation

throughout 60–40 kya, indirectly shedding light on the early interactions and admixture of our species with Neanderthals[33] and the relationships between the main Eurasian and the elusive Basal Eurasian human population[30] as well as informing on where future archaeological investigations should be focused.

## Results

### Rationale

The characteristics of the Hub population were outlined from a genetic perspective, including available data from extant and ancient populations. In consideration of the complex relationship between the Hub population and the EEC and WEC expansions, we aimed to retrieve the genetic profile of the population (Fig. 1A, grey) that remained in the Hub location after the separation from EEC (Fig. 1A, red) which would have shared minimal drift with the WEC (Fig. 1A, blue). In light of patterns of shared drift, we hypothesized that, within a Eurasian landscape, the relic Hub population should be found among those showing a West Eurasian signature, albeit with the least shared drift within representatives of the WEC expansion.

We build a framework by describing ancient and extant populations in terms of their derived allele sharing (DAS) with the two oldest unadmixed representatives of the EEC and WEC waves. The two reference samples that we considered are Tianyuan[27], a 40 ky old individual from East Asia, and Kostenki14[34], a 38 ky old individual from Western Russia. DAS with Tianyuan defines the red, vertical axis (in T units) in Fig. 1B, while DAS with Kostenki14 defines the blue, horizontal axis (in K units). In the absence of admixture and other confounders, most ancient and extant Eurasian genomes should fall either on the red or on the blue axes in a position proportional to the evolutionary time spent together with Tianyuan or with Konstenki14 (R and B in Fig. 1B). Under these ideal circumstances, a population falling at the intersection of the two axes (black dot in Fig. 1B) would mimic the genetic features of the Hub population at the time when the EEC wave departed, while a population falling just right of this point, along the blue axis, would show the legacy of the Hub population after the EEC expansion.

There are, however, at least three confounders that complicate this search: archaic admixtures, interaction between EEC and WEC waves after they left the Hub location (the dashed, blue/red line in Fig. 1A), and admixture with Basal Eurasians (the dashed, blue/green line in Fig. 1A). The genetic contribution from archaic hominins[33,35] decreases as a function of time due to purifying selection[36,37] and is likely to bias the affinities between ancient samples through archaic sharing. This can be overcome by removing sites where Denisovans or Neanderthals share the same derived alleles as the other human populations considered here. On the other hand, groups that experienced additional gene flow from archaic humans (Oceanians and Bacho Kiro)[25] will experience a reduction in their K and T coordinates. The other two confounders are illustrated in Fig. 1A, B and are the result of admixture events that occurred in the last 40 ky. On one hand, an admixture between descendants of the EEC (R, for red, in Fig. 1B) and the WEC (B, for blue, in Fig. 1B) expansions would give birth to an admixed population (BR in Fig. 1B) occupying a position intermediate between the two source populations. On the other hand, the reported admixture of certain groups with the so-called Basal Eurasian population[30] starting at least from 25 kya[31,32] would decrease both their T and K coordinates. Basal Eurasians are a group that is thought to have split from other Eurasians before the divergence between the EEC and the WEC and perhaps even before the establishment of the Hub population. This is, therefore, prior to the intersection of the blue and red axes in Fig. 1B, and is represented as a green dot in the bottom left part of Fig. 1B. Assuming that each analysed genome (Supplementary Data 1–2) was putatively affected by all these confounders, its resulting observed (Obs, Fig. 1B) point should therefore be brought to the B position through a number of corrections before assessing its putative

role as a Hub relic. We also note that a fourth confounder may be represented by a recent interaction with Sub-Saharan genetic components[38].

### DAS with Kostenki/Tianyuan

When examining the positions of modern and ancient populations (Supplementary Data 1–2, Supplementary Fig. 1A) within the reference space defined by the DAS with Kostenki14 and Tianyuan (Fig. 1C, Supplementary Fig. 1B, Supplementary Data 3), it may be observed that East Asians and Pre-Ice Age Europeans resemble the positions held by R and B in Fig. 1B. African populations align along the bisector but before the intersection of the red and blue lines representing $K_T$ (the K coordinate of Tianyuan) and $T_K$ (the T coordinate of Kostenki14), ordered by how early they diverged from the other populations (Khoe and-San are at the bottom, Supplementary Data 3). Oceanians show a greater affinity for Tianyuan compared to Kostenki14, with these populations being predominantly part of the EEC wave[15,18,39]. These populations are, however, shifted towards the bottom left corner of Fig. 1C by their admixture with Denisovans[35], which reduces both K and T values (see Methods for more details on the role of archaic alleles in our calculations). A similar effect can be observed in the Initial Upper Palaeolithic (IUP) Bacho Kiro individuals, which have been shown to have a Neanderthal ancestor in their recent genealogy[25]. As unadmixed South-East Eurasians, the populations from the Andaman Islands are aligned along the red axis at a lower T than East Asians; this is compatible with the shorter time spent together with the ancestors of Tianyuan during the EEC movement away from the Hub location, and with the earliest evidence of human occupation in South Asia at ~48–45 kya[40].

The central portion of the top right quadrant of Fig. 1C is occupied, as expected, by groups resulting from the admixture between the EEC and the WEC waves anytime after they expanded from the Hub location. South Asians follow a cline connecting Andamanese and West Eurasians; this is expected given the reported interaction between Ancestral South Indian (ASI) and Ancestral North Indian (ANI) populations[41,42]. The position of Native Americans suggests a primarily East Asian ancestry, with a smaller contribution from palaeolithic West Eurasian populations[43,44]. Similarly, Mal'ta and Yana fall in an intermediate position between the two axes, the result of a palaeolithic admixture between EEC and WEC groups[18]. The 45 ky old individual from Ust' Ishim[26] falls close to the intersection of the red and blue axes, where we expect the 45 kya Hub population to be positioned. This location is anticipated as the lineage of this individual forms almost a trifurcation with Tianyuan and Kostenki14. Zlatý kůň lies further behind, in accordance with its lineage being unequivocally basal to the split between EEC and WEC populations[18,21]. West Eurasians, North Western South Asians, and Levantines occupy the area below the bisector, compatible with an admixture between EEC and WEC, or below the blue axis, further complicated by the presence of Basal Eurasian or African components in these populations. Since interaction with Sub-Saharan genetic components[38] will have an effect similar to the interaction with Basal Eurasians, we excluded populations showing evidence of gene flow from Africa (Methods, Supplementary Fig. 2, Supplementary Data 4, 5).

### Accounting for DAS confounders

Starting from the K and T coordinates estimated from each ancient or modern genome in Fig. 1C, we assume each assessed genome is potentially affected by Basal Eurasian and/or by admixture between the EEC and the WEC. We exploited the fact that the coordinates of an admixed population correspond to the mean of the two source populations weighted by each source's contribution (we empirically validated this equation by creating synthetic mixed individuals, see Supplementary Data 6–8, Supplementary Fig. 3). For each tested population we assumed the empirically retrieved K and T coordinates

to be $K_{Obs}$ and $T_{Obs}$. On the basis of this, we computed the $K$ coordinate of the source WEC population ($K_B$ in Fig. 1B) following the scheme in Fig. 1B (see Material and Methods). Given that the goal of our analyses was to determine the population with the smallest corresponding $K_B$ coordinate (the one that could fall closest to the Red/Blue origin in Fig. 1), we assessed whether the analytical framework that we developed could retrieve a correct ranking of proximity to the Hub population. Using msprime[45], we performed coalescent simulations to obtain the distinct WEC source populations under different demographic scenarios[46,47] (Methods, Supplementary Code 1–2, Supplementary Fig. 4). In particular, we simulated WEC populations with different allele sharing with Kostenki14 (i.e., mimicking WEC populations with different distances from the Hub population) and mixed them with the EEC and Basal Eurasians (Supplementary Data 9). We found that our approach retrieves the correct ranking along the $K$ coordinate in the majority of cases, and with an accuracy of >0.9 in all cases where the admixed populations are at least 50% WEC, and the mixing WEC sources have at least 3 ky of differential allele sharing with Kostenki14 (Supplementary Fig. 5A). Decreasing the differences between the mixing WEC sources (hence decreasing the differences in their K coordinates) results in higher WEC fractions in the admixed population to maintain a comparable level of accuracy (Supplementary Fig. 5B, C). The lowest accuracy in our simulated results is obtained when the compared populations are the result of the mixture of the WEC and EEC sources (i.e., without any contribution from Basal Eurasians), a composition that appears to be virtually absent in modern Eurasian populations.

Having confirmed the validity of our approach we tested the existing data. We found that after accounting for East and Basal Eurasian confounders, the populations that harbour the WEC component closer to the Hub population (grayscale gradient of population points in Fig. 2A, Supplementary Data 11) are the ones whose West Eurasian ancestry is related to the hunter gatherers and early farmers from Iran[48]. This is a genetic ancestry commonly referred to as the Iran Neolithic[30] or the East Meta[49], here named Iran HG for clarity (Supplementary Data 11). The Iran HG ancestry is widespread not only in modern-day Iran but also across ancient and modern samples from the Caucasus (in particular in the Mesolithic hunter gatherers of that region) and in the northwestern part of South Asia[50]. Along the blue axis of genetic similarity to Kostenki14, these populations come before modern and ancient groups from the Levant and, in turn, before groups from Europe and other areas associated with the Anatolian

Neolithic expansion[49,51–53]. The furthermost groups along this axis are post- and pre-LGM European hunter gatherers, which is expected owing to their genetic proximity to Kostenki14.

We report the affinity to the Hub population of each analysed group showing at least a 75% WEC genetic fraction (the strictest threshold we identified in our validation), after correcting for the aforementioned confounders, in grayscale on the points on a geographic map (Fig. 2A). Assuming at least partial population continuity in the past 40 ky, we identify focal areas[54] for the Hub. These locations are defined as parts of the map where the available West Eurasian genetic component is closest to the Hub population and where such genetic proximity decreases as a function of geographic distance from the said location (Fig. 2A). For each location of the map, with at least one sampled population within 2000 km, we computed the shortest overland distance between all other sampled populations, and we then estimated the Pearson correlation with the corresponding values of $-K_B$. A negative $r$ value (light colours in Fig. 2A) is expected in the locations from where the $K$ coordinates can only increase. The focal area for the Hub location, the lightest shade in Fig. 2A, falls in the region between the southern shores of the Caspian Sea and the Persian Gulf (then not submerged). The results are qualitatively the same when changing the inclusion criteria in terms of the fraction of the West Eurasian genetic component in the admixed individuals (Supplementary Fig. 6A) or the age of the samples (Supplementary Fig. 6B). We performed the same focal area analysis with the inferred proportion of Basal Eurasian (Fig. 2B, Supplementary Fig. 6C, D), and found that the most likely entry point for such an ancestry is separated from the putative Hub location. This appears to be linked to the Levant, suggesting either that the Levant, the Arabian Peninsula or North Africa was a potential location for this elusive population.

In integrating the genetic results within a spatially explicit model, should be noted that post Neolithic expansions might have contributed to the spread of a Hub-like component beyond its homeland; for example, towards northern South Asia, along with the expansion of the so-called Iranian Neolithic genetic components[48,50]. In addition, other population movements might have diluted its presence in the Hub location with the arrival of other WEC components with a lower Hub affinity (e.g., via the Eastward spread of Anatolian Neolithic components)[50]. Therefore, putative legacies of the Hub may be found over a large area, stretching from the Southern Caucasus to northern South Asia, though this may not have always been the case. In the Caucasus, pre-LGM hunter gatherers were more closely related to early

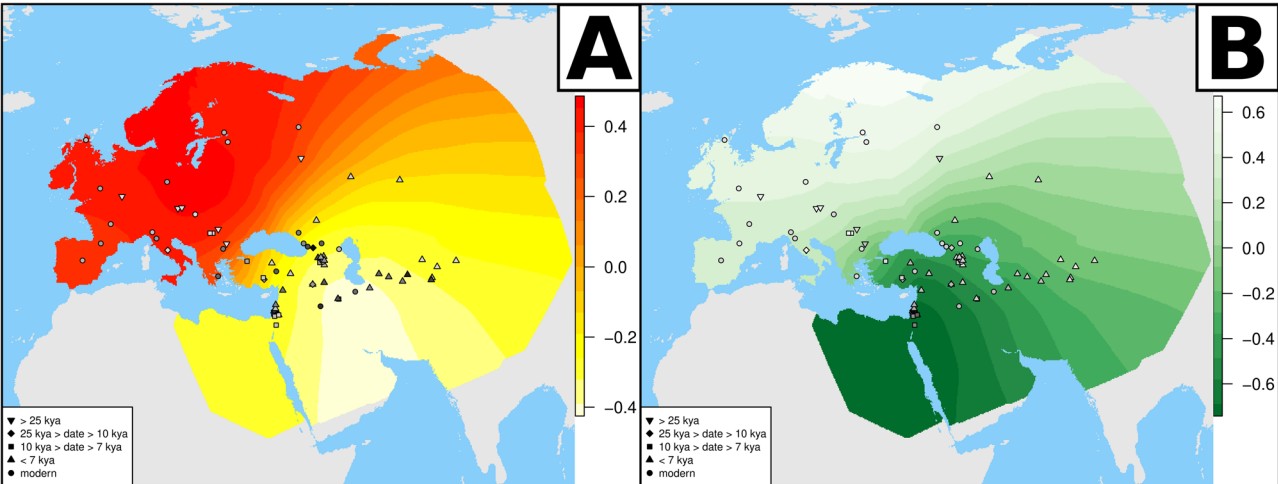

**Fig. 2 | Focal areas for the Hub and Basal Eurasian genetic components.** Focal area for the Hub (**A**, white to light yellow hues show the most likely Hub location from a genetic perspective) and for Basal Eurasian ancestry (**B**, dark hues show higher proportion of a Basal Eurasian component) based on at least 75% WEC ancestry. The grayscale in individual population points is proportional to the inferred proximity to the Hub (**A**) or the proportion of Basal Eurasian ancestry (**B**). kya stands for thousands of years ago. Source Data in Supplementary Data 11.

agriculturalists from western Anatolia[31,32] than to the Mesolithic hunter gatherers (CHG, carrying an ancestry strictly related to Iran HGs). This suggests an expansion of populations from the Hub population to the Caucasus between 25 and 13 kya. This would, therefore exclude the Caucasus as a location for the Hub unless a more complex scenario, such as a double population replacement, is postulated. The presence of a Western Eurasian component in northern South Asia has traditionally been explained as the result of the eastward expansion of Iranian farmers[48]. A recent study, however, reported the presence of this ancestry in a ~4500 year old sample from the Indus Valley, and inferred that it split from Iranian farmers before the advent of agriculture, suggesting that the WEC genetic component may predate the Iranian Neolithic expansion[55]. Nevertheless, as the case of the Caucasus has shown, genetic continuity before the advent of agriculture might not necessarily mean that it dates back to the timeframe of interest. While we can not exclude it, a long term presence of a population Hub in South Asia is at odds with the existence of an indisputably EEC genetic component referred to as ASI (or AASI) that made up the majority of the pre-Neolithic genetic landscape[50].

We inferred the focal area from which Basal Eurasian ancestry expanded into Eurasia, an event reported to have taken place before the Neolithic, but after the WEC expansion. Figure 2B points to northern Africa[56], the Arabian peninsula or the Levant as the most likely homeland for Basal Eurasians, and its expansion across West Asia follows a gradient of radial distance from such areas. Given the inferred focal area for Basal Eurasians, and the notion that such a population was genetically separated from the Hub population until at least 38 kya, it follows that the Hub location must have been physically separated from the Basal Eurasian homeland. For this reason, the putative Hub location should be geographically distinct from the location of Basal Eurasians, a criterion that is met in the Persian Plateau.

## Palaeoecological modelling

The emerging picture for the putative Hub location was based on genetic data alone and reliant on assumptions of population continuity, or at least the absence of major population replacements across the area. To test the plausibility of our continuity assumption, we utilised palaeoecological models as additional and independent layers of information to assess whether such a geographic region was suitable for human occupation at any given time. Recently, palaeoclimatic reconstructions that include the time interval between 70 and 30 thousand years ago have been published[57,58]. We took advantage of this data to build a species distribution model following the procedure described by Rodríguez and colleagues[59] (see Methods, Supplementary Figs. 7–11) and to reconstruct the areas with environmental conditions suitable for human occupation throughout that period (Fig. 3A, Supplementary Fig. 12). We then combined these results with the estimates of Net Primary Productivity (NPP) for each region and the relation between NPP and hunter gatherer population density[59]. This was used to estimate the maximum sustainable human population (carrying capacity) through time in different geographic regions (Fig. 3B, geographic regions shown in Supplementary Fig. 13).

Our palaeoecological model shows that the putative Hub location identified in Fig. 2A (and framed in Fig. 3A) could have supported human occupation throughout most of the time periods between 70 and 30 kya. Our model also shows that the region could have potentially sustained a population size much higher than other Western Asian regions during the same time intervals. This inference can be made even without taking into account the presence of Mesopotamian rivers and the rich hydrological network of the Persian Plateau[60], which likely extended and interconnected areas potentially inhabitable beyond the limits of those deemed suitable by our model. It is interesting to note that the Mesopotamia/Persian Plateau becomes patchier in terms of habitat suitability between 60 and 50 kya (Supplementary Fig. 12); and then reconnects again after 50 kya, a trend also reflected

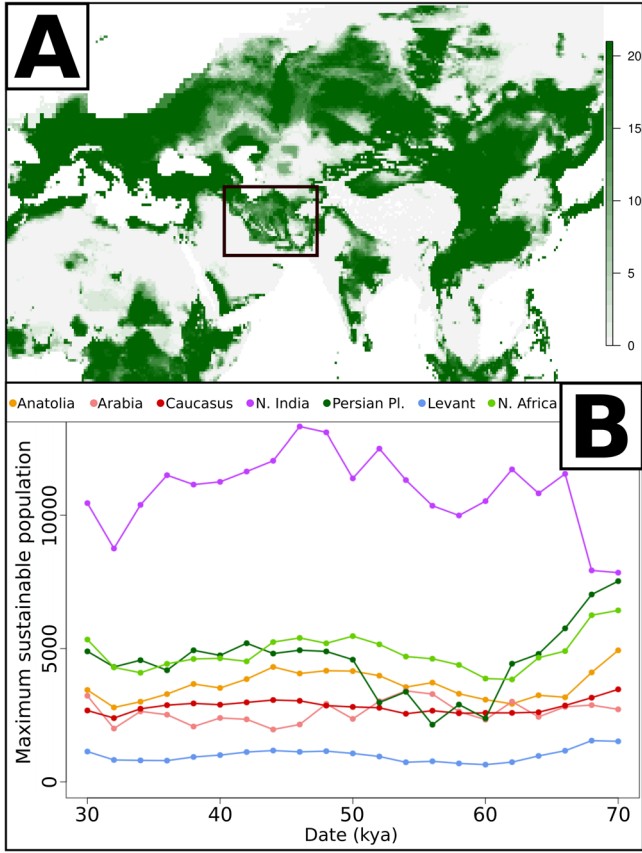

**Fig. 3 | Palaeoecological inference of areas suitable for human occupation.** Number of time intervals between 70 and 30 kya (thousands of years ago, one every 2000 years, 21 in total) in which each area is predicted as suitable for human occupation by our model (**A**). Maximum sustainable human population in each geographic region over time (**B**). The geographic regions are shown in Supplementary Fig. 13. The black frame in **A** shows the area predicted to be the Hub location from genetic evidence. Source Data in Supplementary Data 14.

by the increase in carrying capacity of the region (Fig. 3B). It is possible that this may have provided an ecological trigger for the IUP expansion that occurred sometime before 45 kya[18,61].

Due to the scant availability of direct palaeoclimate records from the Persian Plateau, the data we used for our palaeoecological modelling has been validated locally on a single datapoint from Iran. In spite of this and thanks to the higher resolution of the validating data provided by other areas with similar environmental conditions, the habitat inferred by our model can be considered reliable as independently validated by other approaches. In recent work, researchers utilised paleohydrological mapping along with existing paleoenvironmental proxies and paleoclimate modelling to reconstruct the historical climate of the Persian Plateau, aiming to explore climatically influenced pathways for hominin dispersals during MIS 5 and MIS 3[60]. The paleoclimate model for MIS 3 indicates substantial increases in moisture for both the Zagros Mountains (70–30 kya) and the northern Persian Plateau (50–40 kya) during specific periods within MIS 3. Notably, these conditions could have potentially supported hominin habitation in these areas, a feature also picked up by our model. This aligns with the findings of available proxy records and the spatial distribution of archaeological sites[60,62–67]. When combining this evidence, it appears clear how the ecological variability of the Persian Plateau picked up in our model corresponds with a period of increased aridity during MIS 4 and a later recovery at the onset of MIS 3, which brought more favourable environmental conditions, though not as ideal as those of MIS 5[60]. It is relevant to note that even at its lowest carrying

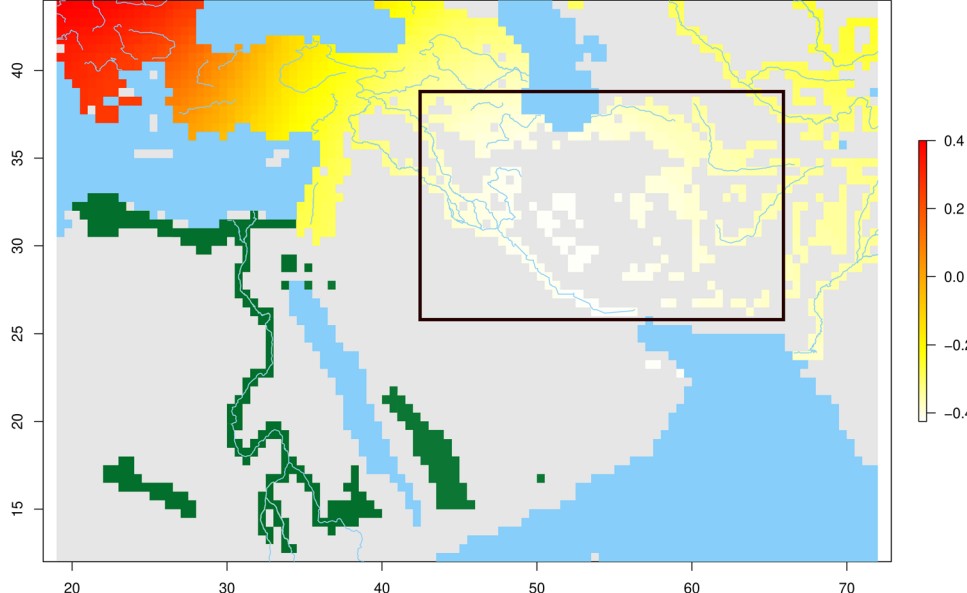

**Fig. 4 | Combination of palaeoecological and genetic analyses.** In light yellow, within the black frame, are geographic locations that are putative Hub focal areas and predicted habitable areas. The areas are compiled on the basis of at least 90% of the time intervals inspected by our palaeoclimatic analysis or those located along major rivers. In green are the habitable areas that might have hosted the Basal Eurasian population.

capacity prediction, the value of the Persian Plateau is almost equal to the highest values in other regions of the Middle East. The Persian Plateau is also higher in comparison to other periods, providing a clue for its competitive advantage over surrounding areas to serve as a Hub location. The palaeoclimatic results, therefore, confirm the suitability of the Persian Plateau for human occupation, validating and refining the picture emerging from the genetic data (Fig. 4, light yellow area). Furthermore, the presence of a viable area located on both shores of the Red Sea and stretching across the Mediterranean Sea would seem to offer a suitable habitat for the Basal Eurasian populations, partly disconnected from the putative Hub location (Fig. 4, green area).

### Fossil and archaeological evidence

The hypothesised links of the EEC wave with the IUP, and of the WEC wave with the Upper Palaeolithic (UP) are informative about the expected material culture to be found in the putative Hub location[18]. According to this scenario, IUP and UP technologies are expected to appear after ~45 kya and ~40 kya, respectively. On the other hand, if the Hub location hosted the population from which even earlier, failed attempts to colonise Eurasia stemmed, such as the assemblage associated with the Zlaty Kun expansion[18,21], cultures perhaps more basal than IUP and UP are also to be expected in the area. With this in mind, and in light of the scarcity of hominin remains in the Persian Plateau dated before 40 kya, a brief examination of the fossil and archaeological record is warranted.

Archaeologists working in the Persian Plateau have begun to pay considerable attention to hominin dispersals in the Late Pleistocene[60,68,69]. A recent climatic and palaeohydrological study has indicated that the region experienced ameliorated conditions in MIS 5 (130–71 kya) and MIS 3 (57–29 kya), consistent with the widespread location of Middle Palaeolithic (MP) sites in the Zagros Mountains and in lowland riverine and lake settings. MP sites in the Zagros Mountains, dating to between ~77–40 kya, are sometimes in association with rare Neanderthal fossils[68,70]. No fossils of *Homo sapiens* have yet been recovered in association with the MP in the Persian Plateau, though associations are found in Arabia at ~85 kya[3] and in the Levant in MIS 5[71] and at ~55 kyr[72]. The latter finding from Manot Cave is dated to 55 ± 5 kya based on the calcitic patina that covered the bone, hence indicating a minimum age[3]. If the actual date was older this individual

would be connected with the former sites, which are too old to represent the ancestors of modern-day non-Africans and are instead associated with an older dispersal OoA. On the other hand, if the dating of Manot reflects its actual age (hence around the time of the OoA), this finding is not at odds with our inference of the Hub being in the Persian Plateau, as the site lies on one of the possible routes leading to it from Africa[13] If this fossil was instead to be considered a member of the Hub population, based on the lack of direct genetic information and given the lack of support for subsequent Levantine individuals to be direct descendants of the Hub population, one needs to postulate a later complete replacement of the hypothetical Levantine Hub population with a less basal lineage, which we here deem to be a non-parsimonious scenario. Archaeologists working in Iran have considered that MP Levallois technologies may also be the product of modern humans[60,73]. And in fact, the MP assemblages of Mirak, dated to ~55 kya, and situated on the Dasht-I Kavir ancient lakes and wetland systems in the Iranian Central Plateau[60,74] have been suggested to represent *Homo sapiens*. This assertion is based on a 3D geometric-morphometric comparison of lithic assemblages, which show the consistency of MP and UP tool reduction strategies at Mirak and their similarity with Levantine assemblages manufactured by modern humans[60,75].

Standardised blade and bladelets industries of the Persian Plateau have been referred to as the Baradostian, the Zagros Aurignancian, and the IUP[68,76,77]. These UP techno-complexes appear to make a relatively rapid appearance across the Persian Plateau[78], in support of a widespread replacement of earlier populations which may have included the Neanderthals and earlier *Homo sapiens* populations[68].

### Discussion

Here we characterised the genetic and geographic features of the Eurasian population Hub, where the ancestors of all present-day non-Africans lived between the early phases of the OoA expansion (~70–60 kya) and the broader colonisation of Eurasia (~45 kya). Our study was not designed to reconstruct earlier population expansions from the Hub[20,21] as well as previous Out of Africa events, which may have left traces in the fossil and archaeological records but contributed little or none of the genetic makeup of present-day individuals[79].

Our results showed that the genetic component closest to the Hub population is represented in ancient and modern populations in

the Persian Plateau. Such a component, after mixing with Basal and East Eurasian ancestries, resurfaced in the palaeogenetic record, previously referred to as the Iranian Neolithic, the Iranian Hunter Gatherer' or the East Meta[49].

This genetic perspective was further refined when combined with palaeoecological evidence, pointing to the Persian Plateau belt surrounding the Central Iranian Desert, and encompassing the Southern Caspian shores, the Zagros Mountains, and the Persian Gulf and Mesopotamia, as the most likely Hub location (Fig. 4). Although no genetic data is currently available, we cannot exclude the northeastern portion of the Arabian peninsula as the southern edge of the Hub[80] since the area was putatively linked across the then drier Persian Gulf. The ecological separation offered by the harsh climatic conditions of the inner parts of present-day southern Levant and western Arabia seems to provide an explanation for the long-lasting disconnect between the Hub population and Basal Eurasians[30], which enabled the two human groups to build their characteristic drift components over many thousand years (their potential locations are shown in green and light yellow colour in Fig. 4).

Information from archaeological evidence advocates for a long presence of modern humans in the region, dated to at least 44 kya when considering the earliest IUP and UP record[68,76,77] or to as early as 55 kya when including certain MP sites that may relate to modern humans[60,75]. Furthermore, the attested presence of Neanderthals in the Zagros mountains until at least 40 kya[81], supports the idea that the ancestors of all living Eurasians spent ~20 ky in the Hub location and admixed with our archaic relatives, potentially over a prolonged period of time[82], ultimately differentiating into the populations that eventually led to the colonisation of Eurasia, Oceania, and the Americas. At around 45 kya, a population expansion associated with the IUP[61,83] emanated from the Hub[18], spreading the EEC ancestry and colonising most of Eurasia[25,84] This expansion left descendants only among present-day East Eurasians[27] and Oceanians, but largely faded and was assimilated in Europe[28] by a subsequent expansion, carried out by WEC groups, taking place ~38 kya[18].

While earlier incursions into Europe have been documented[20,21], they left no trace in the gene pool of present-day and ancient populations. The reasons behind this significant temporal gap between the main OoA event and the first durable expansions deeper into Eurasia can only be hypothesised; but the early stages of the colonisation of Eurasia surely provided with several challenges. A certain amount of time would have been necessary to demographically recover from the OoA bottleneck; and new environmental stressors had to be dealt with either with biological adaptation (see Supplementary Note 1; Supplementary Fig. 14, Supplementary Data)[85] or by developing technological innovations, as testified by the archaeological record. Finally, ecological stressors, exemplified by changes in habitat connectivity and the interaction with local archaic groups might have also played an important role. To this extent, the time elapsed as a single Hub population might have served as an incubator for the development of cultural innovations later observed almost simultaneously at opposite ends of the Pleistocene Eurasian world. Among these features the most notable is the presence of rock art at ~40 kya in Sulawesi, Indonesia[86], compatible in age with the oldest European art (41–35 kya)[87,88]; as well as the innovative usage of projectile weapons, recorded in Europe[89–91], the Levant[92], and South Asia[93].

In conclusion, our multidisciplinary effort shed light on the millennia that separated the Out of Africa expansion and the differentiation of Eurasians into Europeans, East Asians, and Oceanians. Our study also pointed to the Persian Plateau as the most likely candidate for a Hub location of *Homo sapiens* populations, in consideration of genetic, paleontological, and archaeological evidence, thereby illustrating that this is a key region for future archaeological investigations.

## Methods

### Dataset
We downloaded individual genotypes[13,21,25–27,29,30,43,48,50,94–127] for 1,233,013 sites from Allen Ancient_DNA_resource_(version_v.50.0_ https://reich.hms.harvard.edu/allen-ancient-dna-resource-aadr-downloadable-genotypes-present-day-and-ancient-dna-data) and converted it dataset to plink format[128] using admixtools[129] convertf; we then merged it with the Zlatý kůň[21] and Bacho Kiro[25] genomes (processed from bam file), Gumuz[13] genomes and with individuals sequenced in[94]. We selected good quality aDNA samples relevant to our analyses and included all modern-day populations, limiting the number of individuals to 20 (randomly selected) if more than that was available. The full set of individuals analysed is reported in Supplementary Data 1, and a per-population summary is in Supplementary Data 2.

We set the reference allele to the ancestral identified in the 1000 genomes project phase 3 release[123] using plink --reference-allele and kept only autosomes while removing the alleles for which information on the ancestral allele was missing. We also challenged our approach by artificially swapping a fraction of the ancestral/derived allele assignments and found that results are robust even when 50% of alleles are swapped. We used a custom script to identify the derived allele of archaic humans (Neanderthals: Vindija[33], Altai[110], and Chagyrskaya[111]; Denisovans: Altai Denisova[35] and the genome of a first-generation hybrid between the two species[113]) and removed them from the dataset. This left us with 833070 SNPs.

### Derived allele sharing
For each individual or population, using a custom script, we computed the DAS, which is the proportion of derived alleles that each individual shares with a given reference sample) with Kostenki14[34], Tianyuan[27], and Ust'Ishim[26], only considering SNPs with missing call rates not exceeding 25% (using plink --geno 0.25 option).

### Removing individuals with African admixture
To identify the populations showing admixture with Africans we ran an unsupervised admixture[130] analysis on the modern samples of our dataset from $k = 2$ to $k = 10$, choose $k = 7$ (Supplementary Fig. 2, Supplementary Data 4) since it is the one with the lowest cross-validation value and then excluded all populations that show a proportion higher than 0.05 of the ancestry that is maximised in Sub-Saharan African populations (blue in Supplementary Fig. 2). With this step, on top of African populations, we excluded BedouinA and BedouinB, Palestinians, Jordanians, and Yemenite_Yew.

To test whether any of the ancient populations had a contribution from either East or West Africans, we confronted them with French (that were shown by the previous analysis to not have African admixture) by computing the test D(X, French, Yoruba or Gumuz, Chimp) and excluded those with a significantly positive value (Supplementary Data 5).

### Artificially admixed populations
In order to create the chimeric genomes, we first selected three source populations and four individuals from each; we chose Han, Sunghir, and Gumuz as these populations somewhat mimic the R, B, and BEA points in the plot of Fig. 1A. The individuals selected are shown in Supplementary Data 6. We then extracted individual chromosomes for each and merged them as shown in Supplementary Data 7, then computed the proportion of ancestry from each source based on the number of SNPs each chromosome has.

We finally computed the DAS with Kostenki14 and Tianyuan for each source and admixed population (Supplementary Fig. 3, Supplementary Data 8) both empirically (on chimeric genomes) and analytically (by averaging the K and T values of the source populations weighted by their contribution) and compared them. The ratio of the

two values never deviated from 1 more than 0.008 and very often much less than that (Supplementary Data 8).

## Identification of the WEC source population for each sampled population

From this starting point, we made some reasonable assumptions to retrieve the position of the B source population (Fig. 1A) for each Obs (Fig. 1A) population.

The first step is determining the Basal Eurasian contribution in each population. Since Basal Eurasians diverged before the split between EEC and WEC[30], and Ust Ishim forms a near trifurcation with the two above-mentioned groups, all Eurasian populations should have a similar DAS with Ust Ishim ($U$), unless they received some contribution from a more basal population such as Basal Eurasians or an African population (populations displaying a sizeable African contribution, however, were identified and removed from the analyses, see Methods, Supplementary Fig. 2, Supplementary Data 4, 5). We chose as baseline $U$ the mean ($U_{mean}$) of the $U$ of Kostenki14 ($U_K$) and Tianyuan ($U_T$), and since Basal Eurasians have so far not been sampled, we used the $U$, $K$, and $T$ of Gumuz as a conservative proxy for it in Eq. (1) and (2). Gumuz is an East African population that split from Eurasians just before the OOA and, unlike most other East Africans, lack any contribution from Eurasian populations[13]. In this step, if the proportion of Basal Eurasian results is negative, which can be the case for some populations that have a $U$ slightly higher than $U_{mean}$, we set it to zero.

$$pBEA_X = (U_X - U_{mean})/(U_{BEA} - U_{mean}) \qquad (1)$$

Where:

$pBEA_X$ = proportion of Basal Eurasian ancestry in population X

$U_X$ = derived allele sharing with Ust' Ishim of population X

$U_{BEA}$ = derived allele sharing with Ust' Ishim of the Basal Eurasian population

$U_{mean}$ = average of the $U$ of Kostenki14 and Tianyuan

Having estimated the proportion of Basal Eurasian we can then find the position a population would occupy if it did not have it (BR in Fig. 1A).

$$K_{BR} = (K_X - K_{BEA} \cdot pBEA_X)/(1 - pBEA_X) \qquad (2.1)$$

$$T_{BR} = (T_X - T_{BEA} \cdot pBEA_X)/(1 - pBEA_X) \qquad (2.2)$$

Where:

$K_X$, $T_X$ = DAS with Kostenki14, Tianyuan of population X

$K_{BEA}$, $T_{BEA}$ = DAS with Kostenki14 and Tianyuan of the Basal Eurasian population

$K_{BR}$, $T_{BR}$ = $K$ and $T$ coordinates of the hypothetical BR population corresponding to population $X$, i.e., the coordinates it would have if it lacked Basal Eurasian ancestry

We can then determine whether population X has any contribution from an EEC population, knowing that if this was 0 then its $T_{BR}$ coordinate should be equal to $T_{WEC}$; if not, then we need to know the $T$ coordinates of the two sources (B and R in Fig. 1A) in order to determine the proportion of EEC ancestry. We approximate $T_{WEC}$ with the average of the $T$ coordinates of the available WEC populations older than 30 ky (Sunghir[116], Vestonice16[96], Muierii2[109], and Krems[95]).

For $T_{EEC}$, we first need to identify the source EEC population (R in Fig. 1A). Unadmixed EEC populations can be grouped into East Asians (red in Fig. 1B and with higher $T$ values) and Andamanese/ASI (brown in Fig. 1B, with lower T values); to determine which of these two groups was contributing to each population (or if they both did contribute), we used the test D(Irula_S, Han, X, Mbuti). Irula_S are the masked ASI genomes obtained through ancestry deconvolution by Yelmen and colleagues[42], hence a good proxy for an unadmixed South Asian

genetic component. We then used the coordinates of Han as source if the test was significantly negative, the ones of Onge if it was positive and their average if it did not significantly deviate from zero (Supplementary Data 10); we did not use Irula_S and their coordinates directly in our DAS analyses because these individuals were genotyped on a different chip with fewer SNPs than the 1240 K; however ASI has been shown to be a trifurcation with East Asian and Andamanese populations[42]. With this final piece of information, we can compute the proportion of EEC admixture in the hypothetical $BR_X$ point (note that this is different from the proportion in population X if admixture with Basal Eurasians is also present). In this step, if $pEEC_{BRx}$ results are negative or higher than 1, we set it to zero and one, respectively.

$$pEEC_{BR} = (T_{BR} - T_{WEC})/(T_{EEC} - T_{WEC}) \qquad (3)$$

Where:

$pEEC_{BR}$ = the proportion of EEC ancestry in the hypothetical BR population

$T_{WEC}$ = $T$ coordinate of the source WEC population (B in Fig. 1A) of the BR point

$T_{EEC}$ = $T$ coordinate of the source EEC population (R in Fig. 1A) of the BR point

Finally, we compute the coordinates of the unadmixed WEC source (B in Fig. 1A) of each admixed population.

$$K_B = (K_{BR} - K_{EEC} \cdot pEEC_{BR})/(1 - pEEC_{BR}) \qquad (4)$$

Additionally, to get the total proportion of EEC and WEC ancestry in population X, one can compute:

$$pEEC_X = pEEC_{BR} \cdot (1 - pBEA_X) \qquad (5.1)$$

$$pWEC_X = (1 - pEEC_{BR}) \cdot (1 - pBEA_X) \qquad (5.2)$$

Where:

$pEEC_X$ = the proportion of EEC ancestry population X

$pWEC_X$ = the proportion of WEC ancestry population X

## Coalescent simulations

We simulate the OOA 60 kya, with Basal Eurasians (BEA in Supplementary Fig. 4) splitting soon after (57.5 kya) and the split between EEC and WEC, with the former leaving the Hub[18], 46 kya (allowing the time for them to reach Ust'Ishim and Bacho Kiro by ~45 kya). Both Tianyuan and Kostenki14 split off from their respective main branches (EEC and WEC) 34 generations (~1000 years) before their death. We simulated two different West Eurasian populations: WEC and WEC2, with WEC2 staying in the Hub longer than WEC (and Kostenki14), and hence closer to it from a genetic point of view. We then have each of these populations acting as a source for admixture events with Basal Eurasians (BEA) and East Eurasians in different proportions (Supplementary Data 9). We simulated 10 genome chunks for each population and averaged the DAS values obtained from the different chunks of KOS, TIA, and UST populations and then applied the procedure, assumptions, and approximations included, described in the previous paragraph, to identify the B source for each admixed population where WEC is the source and compared it with all other admixed populations where WEC2 is the source and vice versa to see whether, regardless of the ancestry composition of a population, we could retrieve the correct ranking in terms of vicinity to the Hub. We repeated the process in three different scenarios where WEC2 (the source closer to the Hub) shares 3000, 2000, and 1000 years less than WEC with Kostenki14 (i.e., either WEC and WEC2 left the Hub at the same time, but WEC2 branched off earlier or WEC2 remained in the Hub when WEC and

Kostenki14 left), and repeated the process over 500 simulations for each of the three scenarios.

An in-house python3 (reported in Supplementary Code 1) script was used to simulate genomes under the demographic scenario shown in Supplementary Fig. 4A, and the ancestry composition of the admixed populations is shown in Supplementary Data 9. Simulations rely on the msprime library[45] and Ne parameters are taken from the model of Gravel and colleagues[46]. The simulated genomic chunks are written as a vcf file, which is then converted to plink[128]. When comparing the relative position of the inferred sources of WEC ancestry in each admixed population we assigned value 1 if the ranking was correct and 0 if it wasn't, then averaged the matrix over 500 simulations. The resulting matrices are shown in Supplementary Fig. 5A–F.

To test whether our method remained robust under different demographic scenarios, we tested its validity again under the topology proposed by Kamm and colleagues[47], but we slightly modified it by incorporating an initial bottleneck when WEC and EEC populations diverge/leave the Hub and a subsequent exponential growth post Out of Africa with parameters inspired by those inferred by Gravel and colleagues[46]; the model is graphically represented in Supplementary Fig. 4B and the code to generate the simulated genomes is reported in Supplementary Code 2. Finally, since we relied heavily on aDNA data that is largely capture based, we assessed the impact of ascertainment bias using our simulated data: we repeated the analysis for the simulations of the scenario analysed in Supplementary Fig. 5C, restricting the analysis to the SNPs with an minor allele frequency higher than 0.05 in modern Eurasian populations, since the ascertainment process is more likely to identify common variants in largely studied populations. The results do not differ significantly (wilcoxon test $p = 0.9513$, mantel statistic $r = 0.9998 - p = 0.001$ | maximum absolute deviation between two corresponding cells in the two matrices = 0.016); the results are shown in Supplementary Fig. 5G.

## Palaeoecological model

Following the method described in Rodriguez et al.[59] we built a species distribution model (SDM) with presence and background data based on the MaxEnt algorithm[131] to identify environmentally suitable areas for *Homo sapiens* in Eurasia (10°E to 140°W; 0° to 75°N) during the period 70 kya to 30 kya.

We used the georeferenced database of hominin remains and artefacts used in Timmermann et al.[132], which is a derived version of the one published in Raia et al.[133], to fit the SDM. We selected only entries referring to strong evidence (i.e., tier 1, single date in their annotation) of the presence of *Homo sapiens*, dated within the target period (70–30 kya) and area, with dating uncertainty (i.e., difference between maximum and minimum age) lower than 4500 ky.

Estimates for 18 bioclimatic and topographic variables (including temperature and precipitation indices, and elevation (Supplementary Data 12) in the target period with high spatial (0.5° × 0.5°) and temporal (2 kya timesteps) resolution, for the entire area of interest were accessed and extracted using the R package pastclim[57,134]. To avoid biases in fitting the species distribution model using redundant predictors, the 18 bioclimatic and topographic variables were tested for collinearity and multicollinearity on the whole studied area (883260 spatial points). Pairwise Pearson's correlation coefficients (r; Supplementary Fig. 7) were calculated using the R package Hmisc[135], and the predictors strongly correlated with others ($|r| > 0.9$) were excluded from further modelling. The more general predictor was retained when strongly correlated ones occurred (e.g., Annual precipitation over Precipitation of wettest month). Moreover, multicollinearity $R^2$, Tolerance, and Variance Inflation Factor (VIF), all three describing the linear dependence of one predictor on multiple others, were computed using the R package fuzzySim[136]. We iteratively excluded the variable with the highest VIF until all remaining variables matched the condition VIF < 5 as done by Rodriguez and colleagues[59]

(Supplementary Data 13). The candidate variables selected through this process were elevation, minimum annual temperature (BIO5), temperature annual range (BIO7), mean temperature of the wettest trimester (BIO8), precipitation seasonality (BIO15), and precipitation of the driest (BIO17), of the warmest (BIO18) and of the coldest (BIO19) trimesters.

Palaeoclimatic data and georeferenced observations were transformed to the Sphere Two-Point Equidistant projection (ESRI:53031).

Given the unbalanced distribution of sampling effort in the target area, we tested the effect of both sampling size (expressed as a number of considered spatial points) and sampling strategy in selecting the background points required by the MaxEnt algorithm. Absolute sample error at six sample sizes ($n = 1, 10, 100, 1000, 10000, 100000$) was computed for each of the selected predictors, and the procedure was repeated 1000 times at each sample size. Absolute sample error distribution showed to be stable when $n > 1000$, so we decided to use $n = 10,000$ (Supplementary Fig. 8a). The effect of (i) uniformly random sampling over all the time periods, (ii) uniformly random sampling stratified by time period (i.e., weighted by the number of archaeological sites per time period), and (iii) effort-weighted random sampling was tested by comparing predictors distributions at the background points (Supplementary Fig. 8b). For this test we used a sample size of $n = 10,000$. The sampling effort used to weigh background sampling for the third tested strategy was estimated on the collective records of *Homo sapiens* from Timmermann et al.[132] dataset. We selected only records of tier 1 with single date in their annotation with average age smaller than 70 ky. Record points were converted into a continuous surface using a 2D kernel density estimation through the R package ks[137]. We then used the resulting probability surface to randomly sample background points. No considerable differences can be observed between uniform and stratified strategies, while an effect can be observed when using the effort-weighted strategy. Then, we decided to use the effort-weighted strategy to sample background points and avoid biases due to the non-uniform distribution of presence samples over the study area[138].

A total of 54 different configurations of the SDM resulted from the combination of six enhancing functions applied to the feature space, namely L, LQ, LQH, LQP, LQHP, LQHPT (with L = linear, Q = quadratic, H = hinge, P = product, and T = threshold), and nine regularisation multipliers (i.e., 0.2, 0.4, 0.6, 0.8, 1.0, 1.5, 2, 3, and 4). All the SDM configurations were calibrated, and their performance was measured, using the same initial covariates, sample data, and background points.

All SDM configurations were fitted through cross-validation, and then their performance was evaluated considering three indicators: the relative Akaike Information Criterion corrected for small sample sizes (ΔAICc), the average Emission Rate of the 10%-percentile of presence points (OR.10), and the Area Under receiver-operator-Curve (AUC) of the test subsample. None of the SDM configurations tested performed coherently in all three indicators. Models enhanced only by the linear function (L) outperformed in terms of omission rate but performed poorly in ΔAICc and AUC. Opposite results were obtained for the models enhanced by the most complex set of features (LQHPT). Thus, the model with a feature space enhanced by linear, quadratic, and product functions (LQP), and with a regularisation multiplier of 0.2 was selected for further analysis since it represented the best balance among the three indicators (ΔAICc = 978, OR.10 = 0.188, AUC = 0.665; Supplementary Fig. 9).

Within the eight selected predictors, temperature annual range (BIO7), mean temperature of the wettest trimester (BIO8), precipitation seasonality (BIO15), and minimum annual temperature (BIO5) showed to be the most informative ones in estimating human niche (Supplementary Fig. 10).

Then, we predicted habitat suitability for *H. sapiens* in the target area and time periods (Supplementary Fig. 11), producing continuous probability surfaces which were then binarised based on a threshold of

**Article**

suitability equal to 0.238 corresponding to the 5%-percentile of predicted values at observed sites (Supplementary Fig. 12).

Finally, for each geographic region (boundaries shown in Supplementary Fig. 13), we computed the maximum sustainable human population using the estimates of NPP and the regression equations linking NPP to Hunters population density published by Rodriguez and colleagues[59] (Fig. 3B, Supplementary Data 14).

## Reporting summary

Further information on research design is available in the Nature Portfolio Reporting Summary linked to this article.

## Data availability

No new data was produced for this study. Genomic data for modern and ancient individuals is available from the Allen-Ancient DNA Resource (https://reich.hms.harvard.edu/allen-ancient-dna-resource-aadr-downloadable-genotypes-present-day-and-ancient-dna-data). Gumuz genomic data is available from the European Genome-Phenome Archive under accession number EGAS00001000238. Altibathymetric geographic maps of the ETOPO1 dataset[139,140] and the inferred sea levels in the past from the "Global 1 Ma Temperature, Sea Level, and Ice Volume Reconstructions" dataset[141] are available from the NOAA Paleoclimatology Programme. Paleoclimatic Data[57] can be accessed using the R package Pastclim[58]. The data to plot Fig. 1C is reported in Supplementary Data 3. The data to plot Fig. 2 and Supplementary Fig. 6 is reported in Supplementary Data 11 (the coordinates of the populations are reported in Supplementary Data 2). The data to plot Fig. 3B is reported in Supplementary Data 14. Figure 4 combines data from Figs. 2 and 3A, the rivers position was obtained from Natural Earth (free vector and raster map data naturalearthdata.com). The map in Supplementary Fig. 1A is made with Natural Earth, the coordinates of the samples shown are reported in Supplementary Data 2. The data to plot Supplementary Fig. 1B is reported in Supplementary Data 3. The data to plot Supplementary Fig. 2 is reported in Supplementary Data 4. The data to plot Supplementary Fig. 3 is reported in Supplementary Data 8. The data to plot Supplementary Fig. 14 is reported in Supplementary Data 15. Supplementary Fig. 4A, B can be plotted using the code in Supplementary Codes 1 and 2. The code to plot Fig. 3 and Supplementary Figs. 8–12 is available[142].

## Code availability

Supplementary Codes 1 and 2 (the code to run the coalescent simulations) and the custom script used to identify the derived variants in each individual, as well as the code to carry out palaeoecological modeling available on Zenodo (https://zenodo.org/records/10571649)[142].

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

## Acknowledgements

The authors would like to thank Dr. Burak Yelmen for sharing deconvoluted haplotypes of Irula_S genomes. Computation carried out thanks to University of Padova Strategic Research Infrastructure Grant 2017: "CAPRI: Calcolo ad Alte Prestazioni per la Ricerca e l'Innovazione". EB, GM, and SB are funded by ERC-724046-SUCCESS "The earliest migration of Homo sapiens in Southern Europe. Understanding the biocultural processes that define our uniqueness" awarded to Stefano Benazzi. LP is funded by the Italian Ministry of University and Research (PRIN 2022B27XYM).

## Author contributions

Conceived the study: LV, LP. Genomic data analysis: LV, MM, LP. Palaeoecological modelling: CZ, AB, LV. Provided archaeological context: MJS, EB, GM, MDP. Wrote the manuscript: LV, CZ, MJS, MM, MDP, LP. revised the manuscript and provided interpretation of results: EB, GM, SA, TP, SB.

## Competing interests

The authors declare no competing interests
