## [Peer Review File · Nature Communications]

The Persian Plateau served as Hub for Homo sapiens after the main Out of Africa dispersalEditorial Note: This manuscript has been previously reviewed at another journal that is not operating a transparent peer review scheme. This document only contains reviewer comments and rebuttal letters for versions considered at Nature Communications.

Reviewers' Comments:

Reviewer #1:

Remarks to the Author:

I thank Vallini et al. for addressing all my previous comments, as well as those of other reviewers. I continue to think that this is a compelling manuscript, and the evidence presented was made stronger by the revisions the authors made.

The additional testing of robustness to confounding factors such as ascertainment bias, wrongful assignment of ancestral alleles, and the addition of coalescence simulations under the Kamm model, as well as a modified model with the growth parameters, are all very convincing.

I would like to further thank the authors for the addition of Fig 1A, SI Fig 4B, SI Fig 5G.

Yassine Souilmi

Reviewer #2:

Remarks to the Author:

I have reviewed this paper before and focussed mostly on its modifications since. I must say the manuscript is really improved in its current form. The narrative is easier to read and the caveats and problems with the data and assumptions are more frankly acknowledged. Their main finding, about the geographic position and aging of the hub population, might be correct or entirely wrong, but with what we have at hand now and the machinery the authors set up I feel the interpretation is both feasible and correct.

Reviewer #3:

Remarks to the Author:

Vallini et al. present an elegant model to identify the Hub population from which population expansions into Eurasia emerged ~20kya after the Out of Africa (OOA) expansion. The integration of genetic data with palaeoecological modeling identifies the geographic location of the Hub population in the Persian Plateau and provides evidence that the region could have supported human occupation between 70-30kya. The authors developed a novel method to measure derived allele sharing between samples representing the putative ancestral populations of the OOA and subsequent expansions. They use all available high-quality aDNA data and subject the data to rigorous modeling including multiple demographic models, different split times, swapping ancestral/derived allele assignments, and impact of ascertainment bias on the selection of SNPs – their results are robust to these analyses. The authors' results and conclusions are compelling and they provide an excellent summary of the primary human expansions during the critical period between OOA and Eurasia expansions based on genetic and palaeoecological modeling that is also consistent with fossil and archaeological evidence. The figures do an excellent job of explaining the complexity of the different expansions and locations of relevant populations along with their findings.

Reviewer #5:

Remarks to the Author:

This paper presents a combination of genetic evidence and palaeoecological models to infer the geographic region of the Persian Plateau acted as the Hub for our species during the early phases of colonisation of Eurasia. I am not an expert on genetics so will not comment on this part of the paper.

The request was made to comment on the how the authors' have responded to the comments made by Reviewer 4 and these are listed below. I have made 'Additional Comments' on the Authors' responses plus some additional comments for consideration based on the regional archaeology and palaeoenvironments.

Reviewer #4:

I am not an expert in genetics and ecological modelling so will only comment in detail on the other aspects of the paper. This is an interesting paper that has many thought-provoking ideas. However, there are aspects to it that lead me to question the papers conclusions. Given this I feel the paper requires major revisions.

Below I go through the paper from the start commenting on these issues.

R: we would like to thank Reviewer #4 for their constructive suggestions which we have followed or acted upon as described below.

On line 35 it is implied that that the archaeological record of the Persian Plateau supports the genetic findings of the region being a hub. This is not the case as there are no archaeological sites in the region between 75 and 45 ka that can be conclusively attributed to Homo sapiens.

R: following this and another reviewer's suggestion we have now rephrased this sentence as follows "Through palaeoecological modelling we show that the Persian Plateau was suitable for human occupation and that it could sustain a larger population compared to other West Asian regions, strengthening this claim in light of current archaeological evidence." We just want to say that we also explored available archaeological evidence to provide an as comprehensive as possible view on the matter.

Additional comments: Reviewer 4 noted that there are no archaeological sites in the entire region between 75 and 45 ka that can be conclusively attributed to Homo sapiens.

I agree with Reviewer 4 that there are no known sites attributed to Homo sapiens that date between 75 and 45 ka. That is not to say there are no sites but no sites are yet known – so absence of evidence does not necessarily imply evidence for absence (given the size of the Persian/Iranian Plateau at over 3.5 million km²). This is a major weakness in the paper by inferring a 'hub' in the Persian Plateau that lacks supporting dated archaeological evidence and modelled palaeoecological modelling (which again is based on a handful of poorly constrained records over a very large area. See comment later in the review for details relating to the palaeo modelling)

The authors' response here is weak and does not directly address the question posed by Reviewer 4. They do not state the absence of dated sites but comment that that they also explored available archaeological evidence to provide an as comprehensive as possible view on the matter. I am not quite sure what this means? The Abstract has been substantially reworded. On page 1, lines 25-26 they state "The geographic whereabouts of these early settlers in the hiatus between ~70-60 to 45 kya has been difficult to reconcile". There is a reason for this and it is down to no dated evidence!

Reviewer 4. A map is needed in the introduction showing where the sites that are discussed in the text are. They are never clearly shown in any figure. These are the sites where the 3 genomes come from as well as Kostenki14, Sunghir, Bacho Kiro, Tianyuan etc.

R: we now provide a map of the main ancient genomes we explored in our work as Figure S1A.

Additional comments

The authors' provide a new map that clearly shows the sites from where the genomes come from. What is interesting here is the considerable geographic distances shown for these sites compared to the Persian Plateau, which is being presented as the 'hub'. Whilst I appreciate that genomic data is scarce, are the areas too far geographically separated to make the assumptions being presented?

Reviewer 4 stated: Line 130 You should say - This can be overcome by removing sites that are Denisovan or Neanderthal from the searched genetic space.

R: we wanted to keep the mention to the derived status (as opposed to ancestral) of the considered genetic markers. We therefore rephrased as: "This can be overcome by removing sites where Denisova or Neanderthal share the same derived alleles as of the other human populations considered here."

Additional comment: This has been met.

Reviewer 4 stated "Though I am no expert in palaeoecological modelling the fact that just about all of Iran is found to be habitable contradicts what we know about the paleoclimate and thus I find this hard to believe and the discrepancy needs to be explained. The palaeoecological modelling results shown here correctly represents the vast majority of most deserts as being uninhabitable, the exception is the deserts on the Iranian Plateau. How can these deserts be habitable when both paleoclimate models and proxy data show that they always have very low rainfall? Furthermore, this finding contradicts strongly with what is said about the past climate and human habitation in a recent paper on Iran by two of the authors on this paper (Shoaei et al 2023)..... Indeed Shoaei et al (2023), uses the same paleoclimate modelling precipitation data as is applied in the palaeoecological modelling implemented here and the Iranian deserts are always arid and show no evidence for human occupation, even in the more humid periods, and the vast majority of the Iranian Plateaux is only briefly habitable."

R: our results (see Figure S12 for the full extent) use information about aridity and other climatic variables to infer suitability for human occupation. As it can be seen in various panels of Figure S12, the situation in around present-day Iran is most variable, with huge portions of the area switching from habitable to uninhabitable within a matter of few millennia.

We did incorporate these fluctuations in our final claims, by narrowing the putative Hub location down to only those areas that showed a higher stability across the whole time transect covered by our study. This narrower area can be better appreciated in Figure 4, where the light yellow information (coming from the genetic part of our study) forms just a ring around the Iranian Plateau, indeed leaving the central Iranian desert as a non-suitable area.

Additional Comments: The problem with the palaeoecological modelling, as shown in the revised manuscript (page 8, lines 315-364), is that there is such little data available from the Persian Plateau for the time period between 75 and 45 ka. A handful of sites exist and the Persian Plateau is a heterogeneous landscape with considerable variation in terms of relief, rainfall, temperature etc. Actual dated palaeorecords spanning MIS 4 and MIS3 (75 and 45 ka) comes from a single lake record in the northwest (Lake Urmia), a handful of loess/palaeosol records mostly from the foothills of the

Alborz Mountains in the north of the region which are notoriously difficult to date (e.g. do the dates reflect sediment deposition with later pedogenic overprinting or were they contemporaneous?), there are no dated sites in the central region and some poorly dated/constrained terraces exist from the Makran region. The authors need to state/show the evidence used for "We examined published palaeoclimatic data for the time interval between 70 and 30 thousand years ago" as described on page 8, lines 321-322. What were the sites? How many?, evidence used in the models? Somehow, and I am not clear as to how, the authors were able to "reconstruct the areas with environmental conditions suitable for human occupation throughout that period" based on the scant evidence available. (lines 325-326). Whilst some very nice modelled maps are presented I am unsure as to how reliable and robust the limited palaeoenvironmental is for generating such models. This needs to be explained more clearly.

Reviewer 4. As noted above there are no archaeological sites in the region between 75 and ~45 ka that can be definitely attributed to Homo sapiens. The literature attributes sites during this time as belonging to Neanderthals. One reason for this is that the few fossils that have been found in Iran are all Neanderthals. Until a Homo Sapiens fossil is found during the time period of relevance to this study the parsimonious conclusion has to be that the region was occupied by Neanderthals. Given this there is no strong archaeological evidence that the region acted as a hub.

Additional comments.

In the revised manuscript the authors' do cover this on page 11 under the section Fossil and Archaeological Evidence (lines 385-403). Here they state that Middle Palaeolithic sites in the Zagros are sometimes in association with rare Neanderthal fossils and state that no Homo sapiens fossils have been found. The nearest known sites with fossils come from the Levant, and Arabia – so outside the Persian Plateau region. They suggest that archaeologists working in Iran have considered that MP Levallois technologies may also be the product of modern humans but this is tentative in the absence of evidence. MP assemblages of Mirak, dated to ~55 kya, has been suggested to represent Homo sapiens. This is based on the assertion is that comparison of lithic assemblages, which show consistency of MP and UP tool reduction strategies and their similarity with Levantine assemblages manufactured by modern humans. This is paluasive.

A point to note here is that MP archaeology stretches back into MIS5 with no sites dated to between 75 and ~45 ka that can definitely be attributed to Homo sapiens. So the sites could be Neanderthal only or both. This point is made on page 11, lines 408-410.

Color codes to navigate our responses to Reviewer 4 (old) and Reviewer 5 (new):

Black text: Original reviewer #4 comment

Blue text: Our response to reviewer #4

Orange text: Reviewer #5 comment

Purple text: Our response to reviewer #5

This paper presents a combination of genetic evidence and palaeoecological models to infer the geographic region of the Persian Plateau acted as the Hub for our species during the early phases of colonisation of Eurasia. I am not an expert on genetics so will not comment on this part of the paper.

The request was made to comment on the how the authors' have responded to the comments made by Reviewer 4 and these are listed below. I have made 'Additional Comments' on the Authors' responses plus some additional comments for consideration based on the regional archaeology and palaeoenvironments.

We would like to thank Reviewer #5 for their comments on our manuscript and on the replies to Reviewer #4. We recognize our wording might have on some occasions seemed to imply that the fossil and archaeological record was in agreement with our conclusions, that instead mostly rely on our genetic analysis and, to a minor extent, from paleoecological modeling that refined the whereabouts of the Hub Location. We removed the sentence “in light of current archaeological evidence” at the end of the abstract, and now clearly state the scarcity of *Homo sapiens* fossil record in the timeframe of interest at lines 60-62, but highlight that the same is true for all of Eurasia. At lines 83-84 we state that, given the lack of fossil record, we combine genetic and paleoecological models to infer the region that acted as Hub. We agree that given the scarcity of the fossil record 75-45 kya (from the Persian Plateau and from the whole of Eurasia in general) it is not possible to identify the geographic location of the Hub population and hope that our indication of the Persian Plateau as a macro area of interest can be helpful for future archaeological investigations.

I am not an expert in genetics and ecological modelling so will only comment in detail on the other aspects of the paper.

This is an interesting paper that has many thought-provoking ideas. However, there are aspects to it that lead me to question the papers conclusions. Given this I feel the paper requires major revisions.

Below I go through the paper from the start commenting on these issues.

R: we would like to thank Reviewer #4 for their constructive suggestions which we have followed or acted upon as described below.

On line 35 it is implied that that the archaeological record of the Persian Plateau supports the genetic findings of the region being a hub. This is not the case as there are no

archaeological sites in the region between 75 and 45 ka that can be conclusively attributed to *Homo sapiens*.

R: following this and another reviewer's suggestion we have now rephrased this sentence as follows "Through palaeoecological modelling we show that the Persian Plateau was suitable for human occupation and that it could sustain a larger population compared to other West Asian regions, strengthening this claim in light of current archaeological evidence." We just want to say that we also explored available archaeological evidence to provide an as comprehensive as possible view on the matter.

Reviewer 4 noted that there are no archaeological sites in the entire region between 75 and 45 ka that can be conclusively attributed to *Homo sapiens*.

I agree with Reviewer 4 that there are no known sites attributed to *Homo sapiens* that date between 75 and 45 ka. That is not to say there are no sites but no sites are yet known – so absence of evidence does not necessarily imply evidence for absence (given the size of the Persian/Iranian Plateau at over 3.5 million km²). This is a major weakness in the paper by inferring a 'hub' in the Persian Plateau that lacks supporting dated archaeological evidence and modelled palaeoecological modelling (which again is based on a handful of poorly constrained records over a very large area. See comment later in the review for details relating to the palaeo modelling)

The authors' response here is weak and does not directly address the question posed by Reviewer 4. They do not state the absence of dated sites but comment that they also explored available archaeological evidence to provide an as comprehensive as possible view on the matter. I am not quite sure what this means? The Abstract has been substantially reworded. On page 1, lines 25-26 they state "The geographic whereabouts of these early settlers in the hiatus between ~70-60 to 45 kya has been difficult to reconcile". There is a reason for this and it is down to no dated evidence!

We thank the reviewer for the observation. We now state clearly the absence of dated sites in this period at lines 60-62 and 83-84. At line 379-380, we state that there is no *Homo sapiens* fossil in the Iranian Plateau associated with MP. We would like to expand on that and mention that for the timeframe we are studying there is no *Homo sapiens* fossil in all of the Middle East. The only exception might be represented by the cranium from Manot Cave in the Levant (which we mention in the text at line 381), which we now discuss more in detail at lines 381-385. This finding is not at odds with our indication of the Persian Plateau as Hub: at least one of the two most direct routes connecting Africa to Iran (either through the Levant or through Arabia) would have had to be covered (at least transiently) to reach Iran and the date of the Manot Cave findings align perfectly with this scenario. In addition the age estimates come from a calcitic patina or crust that covers the calvaria; hence, these are minimum dates and the specimen might be connected to the older dispersal out of Africa represented by the findings in sites like Skhul and Qafzeh that are too old to represent the dispersal from which modern day Eurasians descend. If we were to consider this single finding as indicative of the Hub being in the Levant only a complete replacement of the Hub population with a less basal lineage could explain the observed genetic results. We now explain this at lines 381-393.

A map is needed in the introduction showing where the sites that are discussed in the text are. They are never clearly shown in any figure. These are the sites where the 3 genomes come from as well as Kostenki14, Sunghir, Bacho Kiro, Tianyuan etc.

R: we now provide a map of the main ancient genomes we explored in our work as Figure S1A.

The authors' provide a new map that clearly shows the sites from where the genomes come from. What is interesting here is the considerable geographic distances shown for these sites compared to the Persian Plateau, which is being presented as the 'hub'. Whilst I appreciate that genomic data is scarce, are the areas too far geographically separated to make the assumptions being presented?

The sites we show in Supplementary Figure 1A shows only the oldest sites in Eurasia from which aDNA data is available that we named and discussed explicitly in the paper (as requested by reviewer #4). The complete list of 1554 genomes we used in our analysis is reported in Supplementary Table 1 (at single individual level) and Supplementary Table 2 (at population level). The geographic position of the populations we ended up using for the genetic inference of the Hub (which were retained based on the results obtained from our simulations) is shown by the grayscale dots in Figure 2 and in Supplementary Figure 6 and do cover the Persian Plateau and all other areas of interest.

On line 56. You say 'After moving into Eurasia, the widespread and stable colonisation of the continent (ca. ~45 kya) occurred through multiple expansions associated with a variety of stone tool technologies 17,18, despite earlier incursions are attested by at least ~54 kya 19–22'. This is a poor English. Furthermore, the 54 ka incursion is crucial to the following sentence that argues for a 20 ka chronological gap despite the evidence at 54 ka. Thus I am not convinced by the 20 ka gap.

R: we didn't want to load this sentence with too many details. The 54 kya event is deemed to have left no descendant across contemporary Eurasians, given that the most ancient European genome sequenced so far (the Zlaty Kun genome, dated to >45kya) is genetically basal to all present day and ancient Eurasians. We therefore consider 20kya as the time elapsed between the Out of Africa range expansion (70-60 kya) and the first expansion out of the Hub that left descendants among present day populations (45 kya, associated with the genetic components found in Tianyuan and across present day East Eurasia).

We have now modified the sentence/paragraph to make it clearer: "The geographically widespread and stable colonisation of Eurasia appears to have occurred at ca. 45 kya through multiple population expansions associated with a variety of stone tool technologies (Vallini et al. 2022; Slimak 2022). Earlier incursions into Europe have been recorded (Slimak et al. 2022; Prüfer et al. 2021; Marciani et al. 2020; Benazzi et al. 2011), however they failed to leave a significant contribution in later populations. A chronological gap of about 20 ky between the Out of Africa migration (~70-60 kya) and the stable colonisation (ca. 45 kya) of West and East Eurasia can be identified, for which the geographic location and genetic features of this population are poorly known."

Line 86. This population, not These population.

R: now corrected

Line 130 You should say - This can be overcome by removing sites that are Denisovan or Neanderthal from the searched genetic space.

R: we wanted to keep the mention to the derived status (as opposed to ancestral) of the considered genetic markers. We therefore rephrased as:
“This can be overcome by removing sites where Denisova or Neanderthal share the same derived alleles as of the other human populations considered here.”

This has been met

Though I am no expert in palaeoecological modelling the fact that just about all of Iran is found to be habitable contradicts what we know about the paleoclimate and thus I find this hard to believe and the discrepancy needs to be explained. The palaeoecological modelling results shown here correctly represents the vast majority of most deserts as being uninhabitable, the exception is the deserts on the Iranian Plateau. How can these deserts be habitable when both paleoclimate models and proxy data show that they always have very low rainfall? Furthermore, this finding contradicts strongly with what is said about the past climate and human habitation in a recent paper on Iran by two of the authors on this paper (Shoaei et al 2023). In Shoaei et al (2023) there is no evidence for occupation of the Iranian deserts and there are only brief windows of time when the climate is found to be suitable for dispersal across Iran by skirting around the edge of these deserts. The only places where there is potential for more long term occupation are the Zagros and Alborz Mountains. Indeed Shoaei et al (2023), uses the same paleoclimate modelling precipitation data as is applied in the palaeoecological modelling implemented here and the Iranian deserts are always arid and show no evidence for human occupation, even in the more humid periods, and the vast majority of the Iranian Plateaux is only briefly habitable.

R: our results (see Figure S12 for the full extent) use information about aridity and other climatic variables to infer suitability for human occupation. As it can be seen in various panels of Figure S12, the situation in around present-day Iran is most variable, with huge portions of the area switching from habitable to inhabitable within a matter of few millennia. We did incorporate these fluctuations in our final claims, by narrowing the putative Hub location down to only those areas that showed a higher stability across the whole time-transect covered by our study. This narrower area can be better appreciated in Figure 4, where the light yellow information (coming from the genetic part of our study) forms just a ring around the Iranian Plateau, indeed leaving the central Iranian desert as a non-suitable area.

The problem with the palaeoecological modelling, as shown in the revised manuscript (page 8, lines 315-364), is that there is such little data available from the Persian Plateau for the time period between 75 and 45 ka. A handful of sites exist and the Persian Plateau is a heterogeneous landscape with considerable variation in terms of relief, rainfall, temperature

etc. Actual dated palaeorecords spanning MIS 4 and MIS3 (75 and 45 ka) comes from a single lake record in the northwest (Lake Urmia), a handful of loess/palaeosol records mostly from the foothills of the Alborz Mountains in the north of the region which are notoriously difficult to date (e.g. do the dates reflect sediment deposition with later pedogenic overprinting or were they contemporaneous?), there are no dated sites in the central region and some poorly dated/constrained terraces exist from the Makran region. The authors need to state/show the evidence used for “We examined published palaeoclimatic data for the time interval between 70 and 30 thousand years ago” as described on page 8, lines 321-322. What were the sites? How many?, evidence used in the models? Somehow, and I am not clear as to how, the authors were able to “reconstruct the areas with environmental conditions suitable for human occupation throughout that period” based on the scant evidence available. (lines 325-326). Whilst some very nice modelled maps are presented I am unsure as to how reliable and robust the limited palaeoenvironmental is for generating such models. This needs to be explained more clearly.

The paleoclimatic data we used for ecological modeling was made available by Beyer and colleagues in their 2020 paper “High-resolution terrestrial climate, bioclimate and vegetation for the last 120,000 years” (<https://www.nature.com/articles/s41597-020-0552-1>), and is accessible through the R package `pastclim` (<https://github.com/EvolEcolGroup/pastclim>). These inferred palaeoclimatic parameters have been successfully validated against geographically and temporally localised empirical climate reconstruction and measurements (which we believe may include also the Lake Urmia site mentioned by the reviewer), and have been widely used by researchers receiving 40 citations in 3 years. The dataset includes 19 variables that are reported in Supplementary Table 11 (we also copied the table below this reply for ease of consultation). The paleoclimatic reconstruction has a geographic resolution of 0.5° of latitude/longitude and, for the timeframe of interest, with records every 2000 years. As requested by the reviewer we have added a more detailed description of the dataset we used on line 310-312.

Abbreviation	Description
BIO1	Annual mean temperature
BIO4	Temperature seasonality
BIO5	Minimum annual temperature
BIO6	Maximum annual temperature
BIO7	Temperature annual range
BIO8	Mean temperature of the wettest quarter
BIO9	Mean temperature of the driest quarter
BIO10	Mean temperature of the warmest quarter
BIO11	Mean temperature of the coldest quarter

BIO12	Annual precipitation
BIO13	Precipitation of the wettest month
BIO14	Precipitation of the driest month
BIO15	Precipitation seasonality
BIO16	Precipitation of wettest quarter
BIO17	Precipitation of driest quarter
BIO18	Precipitation of warmest quarter
BIO19	Precipitation of coldest quarter
NPP	Net primary productivity
Elevation	Altitude

Reviewer 4. As noted above there are no archaeological sites in the region between 75 and ~45 ka that can be definitely attributed to *Homo sapiens*. The literature attributes sites during this time as belonging to Neanderthals. One reason for this is that the few fossils that have been found in Iran are all Neanderthals. Until a *Homo Sapiens* fossil is found during the time period of relevance to this study the parsimonious conclusion has to be that the region was occupied by Neanderthals. Given this there is no strong archaeological evidence that the region acted as a hub.

We understand the reviewer's observation, however, fossil records attributable to *Homo sapiens* in the timeframe of interest are lacking everywhere and not just in the Persian Plateau, since the sites from Arabia and the Levant are too old to be representative of the OoA from which modern non Africans descend. The only exception might be the single cranium recovered from Manot Cave. As mentioned in the reply to the first comment of Reviewer #5, however, this specimen might be older (and hence connected to the aforementioned earlier OoA), and even if correct is not at odds with the Persian Plateau being the Hub since it lies on the route connecting Africa to it.

So following the parsimonious explanation would lead to saying that *Homo sapiens* was not in Eurasia at all, however we know that *Homo sapiens* was indeed out of Africa during this time as the hybridization with Neanderthals occurred at around 55 kya.

We now argue that precisely for this mismatch between *Homo sapiens* fossils available to date and genetic inference made for the dynamics of the Out of Africa range expansion, this time and area needs to be better investigated, and hope that our work would help in such effort by indicating a macro area of potential interest.

As noted above there are no archaeological sites in the region between 75 and ~45 ka that can be definitely attributed to *Homo sapiens*. The literature attributes sites during this time as

belonging to Neanderthals. One reason for this is that the few fossils that have been found in Iran are all Neanderthals. Until a Homo Sapiens fossil is found during the time period of relevance to this study the parsimonious conclusion has to be that the region was occupied by Neanderthals. Given this there is no strong archaeological evidence that the region acted as a hub.

R: We agree with the reviewer about the fact that none of the Middle Palaeolithic (MP) stone tools found in Iran were convincingly associated with *Homo sapiens* and that, out of parsimony, they were instead attributed to Neanderthal. At the same time, however, the idea of MP technologies are an exclusive signature of a single human species is outdated (Greenbaum et al. 2019), and Neanderthals and sapiens must have lived in sympatry in some areas, as testified by the interbreeding between the two species (Green et al. 2010). In this light, the MP sites found between the central Iranian desert and the southern Caspian shores display attributes that are markedly different from the MP sites on the Zagros Mountains. For this reason, we speculate that such sites may, in perspective, be seen as a trace of some other population (e.g. *H. sapiens*) in the area. We do agree that this cannot be seen as a strong archaeological evidence (we have indeed rephrased the abstract) that the region acted as a hub. Our commentary on the archaeological records should be seen more as an attempt to stress how our results may help put the available and yet to come archaeological evidence under a novel perspective.

We have now also rephrased the relevant section in the manuscript (along with many other places, following a general re-styling of our text) and added further references to make our point clearer:

“No fossils of *Homo sapiens* have yet been recovered in association with the MP in the Persian Plateau, though associations are found in the Levant in MIS 5 (Shea 2008) and at ~55 kyr (Hershkovitz et al. 2015) and in Arabia at ~85 kyr (Groucutt et al. 2018). Given this, archaeologists working in Iran have considered that MP Levallois technologies may also be the product of modern humans (S. Heydari-Guran and Ghasidian 2021; Shoaee et al. 2023). And in fact the MP assemblages of Mirak, dated to ~55 kya, and situated on the Dasht-e Kavir ancient lakes and wetland systems in the Iranian Central Plateau (Nasab et al. 2019; Shoaee et al. 2023) has been suggested to represent *Homo sapiens*. This assertion is based on a 3D geometric-morphometric comparison of lithic assemblages, which show consistency of MP and UP tool reduction strategies at Mirak and their similarity with Levantine assemblages manufactured by modern humans (Hashemi et al. 2021; Shoaee et al. 2023).

Standardised blade and bladelets industries of the Persian Plateau have been referred to as the Baradostian, the Zagros Aurignacian and the IUP (Shidrang 2018; Ghasidian 2019; Shoaee, Vahdati Nasab, and Petraglia 2021). These UP techno-complexes appear to make a relatively rapid appearance across the Persian Plateau (Ghasidian, Heydari-Guran, and Mirazón Lahr 2019), in support of a widespread replacement of earlier populations which may have included the Neanderthals and earlier *Homo sapiens* populations (Shoaee, Vahdati Nasab, and Petraglia 2021).”.

In the revised manuscript the authors' do cover this on page 11 under the section Fossil and Archaeological Evidence (lines 385-403). Here they state that Middle Palaeolithic sites in the Zagros are sometimes in association with rare Neanderthal fossils and state that no *Homo sapiens* fossils have been found. The nearest known sites with fossils come from the Levant,

and Arabia – so outside the Persian Plateau region. They suggest that archaeologists working in Iran have considered that MP Levallois technologies may also be the product of modern humans but this is tentative in the absence of evidence. MP assemblages of Mirak, dated to ~55 kya, has been suggested to represent *Homo sapiens*. This is based on the assertion is that comparison of lithic assemblages, which show consistency of MP and UP tool reduction strategies and their similarity with Levantine assemblages manufactured by modern humans. This is plausible.

A point to note here is that MP archaeology stretches back into MIS5 with no sites dated to between 75 and ~45 ka that can definitely be attributed to *Homo sapiens*. So the sites could be Neanderthal only or both. This point is made on page 11, lines 408-410.

We would like to thank Reviewer 5 for their overall appreciation for our careful wording when hypothesizing a link between the later MP sites in Iran and a putative presence of *Homo sapiens*.

SI

Supplementary table 11 and 12 do not exist.

R: Supplementary tables up to S13 were present in the file originally submitted, but they might have been hidden by the Excel default visualization of multiple tabs. We did however notice that the Table numbers in the caption of Tables S11-13 were actually reported as tables 10-12, which may have added to the confusion. Now all is in order.

We also would like to report that the revised manuscript also incorporates the suggestion concerning a missing space and a reference formatting made as an attached pdf document.

Reviewers' Comments:

Reviewer #5:

Remarks to the Author:

I should like to thank the authors for the prompt response to comments made by this reviewer. Most of the questions/queries have been addressed and reworded more carefully so not to be front loading the paper as being 'the hub' for later dispersals into Asia and Europe.

The only area that I am still not in agreement with and needs to be reconsidered/reevaluated is related to the palaeoecological modelling.

Original question posed by this reviewer was: "The problem with the palaeoecological modelling, as shown in the revised manuscript (page 8, lines 315-364), is that there is such little data available from the Persian Plateau for the time period between 75 and 45 ka. A handful of sites exist and the Persian Plateau is a heterogeneous landscape with considerable variation in terms of relief, rainfall, temperature etc. Actual dated palaeorecords spanning MIS 4 and MIS3 (75 and 45 ka) comes from a single lake record in the northwest (Lake Urmia), a handful of loess/palaeosol records mostly from the foothills of the Alborz Mountains in the north of the region which are notoriously difficult to date (e.g. do the dates reflect sediment deposition with later pedogenic overprinting or were they contemporaneous?), there are no dated sites in the central region and some poorly dated/constrained terraces exist from the Makran region. The authors need to state/show the evidence used for "We examined published palaeoclimatic data for the time interval between 70 and 30 thousand years ago" as described on page 8, lines 321- 322. What were the sites? How many?, evidence used in the models? Somehow, and I am not clear as to how, the authors were able to "reconstruct the areas with environmental conditions suitable for human occupation throughout that period" based on the scant evidence available. (lines 325-326). Whilst some very nice modelled maps are presented I am unsure as to how reliable and robust the limited palaeoenvironmental is for generating such models. This needs to be explained more clearly.

The authors response was:

The paleoclimatic data we used for ecological modeling was made available by Beyer and colleagues in their 2020 paper "High-resolution terrestrial climate, bioclimate and vegetation for the last 120,000 years" (<https://www.nature.com/articles/s41597-020-0552-1>), and is accessible through the R package pastclim (<https://github.com/EvolEcolGroup/pastclim>). These inferred palaeoclimatic parameters have been successfully validated against geographically and temporally localised empirical climate reconstruction and measurements (which we believe may include also the Lake Urmia site mentioned by the reviewer), and have been widely used by researchers receiving 40 citations in 3 years.

The dataset includes 19 variables that are reported in Supplementary Table 11 (we also copied the table below this reply for ease of consultation). The paleoclimatic reconstruction has a geographic resolution of 0.5° of latitude/longitude and, for the timeframe of interest, with records every 2000 years.

As requested by the reviewer we have added a more detailed description of the dataset we used on line 310-312.

Reviewers new response.

I am still not entirely convinced with the authors reply. For the entire Persian Plateau region only one site is included in the Beyer et al 2020 paper data set (Lake Urmia) which is on the periphery.

Therefore almost the entire region under consideration has NO palaeo records that cover the time span under consideration. Therefore the model used, even though it is considered high resolution is pure modelled interpolation based on a minimal/low resolution of site. This need to be pointed out that it is a model based on only one data point. It may be high resolution in terms of the model applied but the temporal and spatial number of sites underpinning this modelling is not. The authors need to revisit this as the connotations being presented may be overstated.

Reviewer #5 (Remarks to the Author):

Reviewers new response.

I am still not entirely convinced with the authors reply. For the entire Persian Plateau region only one site is included in the Beyer et al 2020 paper data set (Lake Urmia) which is on the periphery. Therefore almost the entire region under consideration has NO palaeo records that cover the time span under consideration. Therefore the model used, even though it is considered high resolution is pure modelled interpolation based on a minimal/low resolution of site. This need to be pointed out that it is a model based on only one data point. It may be high resolution in terms of the model applied but the temporal and spatial number of sites underpinning this modelling is not. The authors need to revisit this as the connotations being presented may be overstated.

R: Following the reviewer's suggestion, we have now extensively reworded the *Palaeoecological modelling* paragraph to fully disclose the availability of a single datapoint for Iran to validate the palaeoclimatic data underlying our ecological model. Nevertheless we also point out that the outcome of our model was shown to be predictive when compared with other types of evidence (e.g. paleohydrological mapping).

Here is our reworded paragraph:

“Due to the scant availability of direct palaeoclimate records from the Persian Plateau, the palaeoclimatological data we used for our paleoecological modeling has been validated locally on a single datapoint from Iran. In spite of this and thanks to the higher resolution of the validating data provided by other areas with similar environmental conditions, the habitat inferred by our model can be considered reliable as independently validated by other approaches. In a recent work, researchers utilised paleohydrological mapping along with existing paleoenvironmental proxies and paleoclimate modelling to reconstruct the historical climate of the Persian Plateau, aiming to explore climatically influenced pathways for hominin dispersals during MIS 5 and MIS 3. The paleoclimate model for MIS 3 indicates substantial increases in moisture for both the Zagros Mountains (70-30 kya) and the northern Persian Plateau (50-40 kya) during specific periods within MIS 3. Notably, these conditions could have potentially supported hominin habitation in these areas, a feature also picked up by our model. This aligns with the findings of available proxy records and the spatial distribution of archaeological sites. When combining these evidence, it appears clear how the ecological variability of the Persian Plateau picked up in our model corresponds with a period of increased aridity during MIS 4 and a later recovery at the onset of MIS 3, which brought more favourable environmental conditions, though not as ideal as those of MIS 5.”